# Transcriptional differentiation of *Trypanosoma brucei* during *in vitro* acquisition of resistance to acoziborole

Pieter C. Steketee[1], Federica Giordani[2], Isabel M. Vincent[2], Kathryn Crouch[2], Fiona Achcar[2], Nicholas J. Dickens[2], Liam J. Morrison[1], Annette MacLeod[2], Michael P. Barrett[2,3]*

1 The Roslin Institute, Royal (Dick) School of Veterinary Studies, University of Edinburgh, Easter Bush, Midlothian, United Kingdom, 2 Wellcome Centre for Integrative Parasitology, Institute of Infection, Immunity and Inflammation, College of Medical, Veterinary and Life Sciences, University of Glasgow, United Kingdom, 3 Glasgow Polyomics, University of Glasgow, United Kingdom

* Michael.Barrett@glasgow.ac.uk

## Abstract

Subspecies of the protozoan parasite *Trypanosoma brucei* are the causative agents of Human African Trypanosomiasis (HAT), a debilitating neglected tropical disease prevalent across sub-Saharan Africa. HAT case numbers have steadily decreased since the start of the century, and sustainable elimination of one form of the disease is in sight. However, key to this is the development of novel drugs to combat the disease. Acoziborole is a recently developed benzoxaborole, currently in advanced clinical trials, for treatment of stage 1 and stage 2 HAT. Importantly, acoziborole is orally bioavailable, and curative with one dose. Recent studies have made significant progress in determining the molecular mode of action of acoziborole. However, less is known about the potential mechanisms leading to acoziborole resistance in trypanosomes. In this study, an *in vitro*-derived acoziborole-resistant cell line was generated and characterised. The Aco[R] line exhibited significant cross-resistance with the methyltransferase inhibitor sinefungin as well as hypersensitisation to known trypanocides. Interestingly, transcriptomics analysis of Aco[R] cells indicated the parasites had obtained a procyclic- or stumpy-like transcriptome profile, with upregulation of procyclin surface proteins as well as differential regulation of key metabolic genes known to be expressed in a life cycle-specific manner, even in the absence of major morphological changes. However, no changes were observed in transcripts encoding CPSF3, the recently identified protein target of acoziborole. The results suggest that generation of resistance to this novel compound *in vitro* can be accompanied by transcriptomic switches resembling a procyclic- or stumpy-type phenotype.

## Author summary

Human African Trypanosomiasis (HAT), also known as Sleeping Sickness, is a parasitic disease prevalent across sub-Saharan Africa, where it is transmitted by blood-sucking

**Data Availability Statement:** RNAseq data is available at GEO (Accession number: GSE168394).

Metabolomics data is available at Metabolights (Accession number: MTBLS2559).

**Funding:** This work was funded by a Wellcome Trust (https://wellcome.org/) PhD studentship awarded to PCS (096980/Z/11/Z). In addition, this work was funded by an MRC (https://mrc.ukri.org/) award to MPB (MR/K008749/1) and a Wellcome Trust core grant to the Wellcome Centre for Integrative Parasitology (grant 104111/Z/14/Z). The Roslin Institute is core funded by the BBSRC (https://bbsrc.ukri.org/; BS/E/D/20002173). The funders had no role in study design, data collection and analysis, decision to publish, or preparation of the manuscript.

**Competing interests:** The authors have declared that no competing interests exist.

Tsetse flies. The disease is caused by a single-celled protozoan parasite called *Trypanosoma brucei*. HAT case numbers have decreased over the past two decades, and elimination is in sight. However, to meet the WHO objective of sustainable HAT elimination, newer and safer drugs are needed. Acoziborole is a new drug that has proven safe and effective against HAT and, unlike previous drugs, it can be orally administered. Whilst the mode of action of acoziborole was recently identified, less is known regarding the mechanisms by which *T. brucei* could become resistant to acoziborole. To address this, we generated a drug resistant cell line under laboratory conditions, in order to analyse the differences between resistant and sensitive cells. By sequencing the transcriptome, we observed that many genes associated with mammalian-infective parasites are down-regulated, whilst genes associated with the insect stage are up-regulated. As a result of these differences, the metabolic effects of acoziborole on parasites are nullified. These data suggest that parasite differentiation, albeit on a gene expression level, is potentially a mechanism of drug resistance in *T. brucei*, although it is doubtful whether this mechanism could occur in the field where mammalian immune effectors would destroy parasites differentiating this way.

## Introduction

Human African Trypanosomiasis (HAT; also known as Sleeping Sickness) is a disease endemic to sub-Saharan Africa, caused by subspecies of the parasitic protozoan *Trypanosoma brucei*. In addition, *T. brucei brucei* and *T. brucei rhodesiense* are infectious to livestock, causing Animal African Trypanosomiasis (AAT) [1]. Together, HAT and AAT account for a significant socio-economic burden across the sub-Saharan African continent. *T. b. gambiense* accounts for the majority (~98%) of HAT cases, and is endemic in 24 countries in west and central Africa [2], whilst *T. b. rhodesiense* causes the remaining infections and is restricted to east and southern Africa [3]. Whilst the rate of disease progression differs in duration between the two species, both begin with an early haemo-lymphatic stage of infection (stage 1), followed by a late-stage (neurological stage; stage 2), and without treatment, the disease is usually fatal [4].

African trypanosomes display a complex life cycle that includes procyclic (PCF) stages in the tsetse fly as well as a mammalian-infective bloodstream (BSF) stage. Pleomorphic *T. brucei* strains are rapidly dividing in the BSF stage, with a long and slender morphology, but possess the ability to differentiate into non-dividing short stumpy forms, which are pre-adapted to survival in the insect vector [5]. This developmental transition is triggered by a quorum sensing mechanism involving a "stumpy induction factor", recently identified as oligopeptide uptake mediated by the TbGPR89 transporter [6]. Differentiation between the various life cycle stages is accompanied by morphological changes, as well as global shifts in transcriptome and metabolome (including downregulation of BSF-specific variant surface glycoproteins - VSGs - and associated expression sites) [5,7–12]. Characterisation of the *T. brucei* life cycle stages has yielded the identification of several stage-specific markers. For example, stumpy-form *T. brucei* exhibits upregulation of transcripts encoding PAD (protein associated with differentiation) [9], as well as downregulation of histones, DNA replication/repair and cytoskeleton-related transcripts, indicative of their non-proliferative state [7,13]. In addition, both stumpy forms and PCFs express transcripts encoding EP and GPEET procyclins in place of VSGs [14]. Differentiation to stumpy form and PCF also involves downregulation of transcripts associated with glycolysis, as the parasite begins to shift to mitochondrial metabolism [8,12].

Sustained efforts to control HAT have led to a significant reduction in *T. b. gambiense* case number. Whilst hundreds of thousands of cases were estimated at the millennium [15], only

992 were reported in 2019 [16]. Elimination by interruption of transmission of HAT is therefore a 2030 target in the WHO Neglected Tropical Disease initiative [17]. However, to realise this target for *T. b. gambiense*, and to be able to more efficiently treat cases of *T. b. rhodesiense*, safer drugs, preferably with oral bioavailability are needed to combat the disease. Historically, few chemotherapeutics have been available to treat HAT. Furthermore, some of those that are used currently are species (eflornithine) and disease stage-specific, with notable issues in administration routes, toxicity and emerging resistance [18–22]. The primary drugs used for stage 1 HAT are pentamidine and suramin, whilst melarsoprol, eflornithine and nifurtimox (the latter two now usually as a combination therapy against *T. b. gambiense*) are used to treat stage 2 infections [23]. None of these drugs are orally bioavailable and they involve long treatment regimens [24].

Recent efforts have led to the development of two promising candidates, fexinidazole, which has now been licensed for use, but requires protracted 10 day administration [25], and acoziborole (AN5568 or SCYX-7158), a benzoxaborole currently in phase IIb/III clinical trials [18]. Benzoxaboroles are a relatively novel class of compounds, characterised by a boron-heterocyclic scaffold, which exhibits a broad spectrum of medicinal applications including antifungal [26], anti-parasitic [27–29], anti-viral [30] and anti-bacterial [31] activity.

Several studies have shed light on the mode of action of acoziborole, as well as related benzoxaboroles. For example, the protein target of the anti-bacterial tavaborole (AN2690) was shown to be leucyl-tRNA synthetase [26]. Recently, a target of acoziborole and the livestock trypanocide AN11736, and also other benzoxaboroles [32], was identified as the trypanosome Cleavage and Polyadenylation Specificity Factor 3 (CPSF3), a nuclear mRNA processing endonuclease [33]. The same target was identified for the benzoxaborole AN3661, active against *Plasmodium falciparum* [34]. Our own studies showed that acoziborole treatment leads to significant perturbations in S-adenosyl-L-methionine metabolism in *T. brucei*, as well as further metabolic phenotypes similar to those elicited by the methyltransferase inhibitor sinefungin [35].

In addition to mode of action, investigating potential mechanisms of drug resistance is crucial in optimising drug application in the field and prolonging the life time of chemotherapeutics [36,37]. Generation of resistance in an *in vitro* setting, when possible, has proven effective in predicting mechanisms of naturally acquired resistance in the field [38]. One previous study investigated mechanisms of trypanosome resistance against acoziborole, but the results were inconclusive, although amplification of the CPSF3 locus was observed in at least one acoziborole-resistant line [39]. In the case of the valyl-ester containing compound AN11736, which was in development for AAT [29], resistance has been shown to be due to loss of a protease that cleaves the parent prodrug to a carboxylate derivative that then accumulates to high levels leading to high levels of potency [40]. A similar prodrug cleavage mechanism was described for another trypanocidal benzoxaborole [41].

In this study, the generation of an *in vitro T. brucei* cell line with high levels of resistance to acoziborole is presented. This cell line exhibited cross-resistance, as well as hypersensitivity, to several trypanocides, including increased resistance to sinefungin. Transcriptomics analysis of the resistant cell line revealed global upregulation of stumpy- or procyclic-specific genes and downregulation of BSF-specific genes, giving the appearance of a "stumpy" or "procyclic" transcriptome. No changes were observed in CPSF3 transcript abundance. However, the acoziborole-resistant cells retained BSF morphology. Analysis of the metabolome showed no changes in S-adenosyl-L-methionine levels in resistant cells treated with acoziborole, indicating that the metabolic phenotype of acoziborole treatment was abolished. In summary, acoziborole resistance, *in vitro*, appears to be characterised by global transcriptomic changes, leading to a partial switch towards procyclic mRNA abundances in resistant cells.

## Results

### *In vitro* selection of acoziborole-resistance

An acoziborole-resistant (Aco[R]) *T. brucei* cell line was generated by continuous *in vitro* culture of the Lister 427 strain. Notably, this strain is known to be monomorphic, and is incapable of differentiation through normal routes to procyclics via stumpy forms–including a failure to trigger mitochondrial oxidative phosphorylation [42,43]. Cells were cultured in the presence of incremental increasing doses of the benzoxaborole, starting at 170 nM (Fig 1). A wild-type cell line was also maintained in the absence of acoziborole to account for culture medium-derived artefacts or adaptation. After ~200 days, Aco[R] cells were viable in 4.96 μM of the compound (Fig 1A) and at this point, were deemed to be resistant compared to the parental control. At this point, the cell line was cloned by dilution for downstream experiments and four resistant clones were isolated. Cell doubling time was noticeably increased in all four clones, compared to a wild-type control (7.3 ± 0.43 h and 14.92 ± 3.19 h for wild-type and Aco[R] cells, respectively; Fig 1B).

To confirm increased resistance against the benzoxaborole, alamar blue assays were carried out in order to generate EC$_{50}$ values. The Aco[R] line exhibited a 15-20-fold increase in EC$_{50}$ to

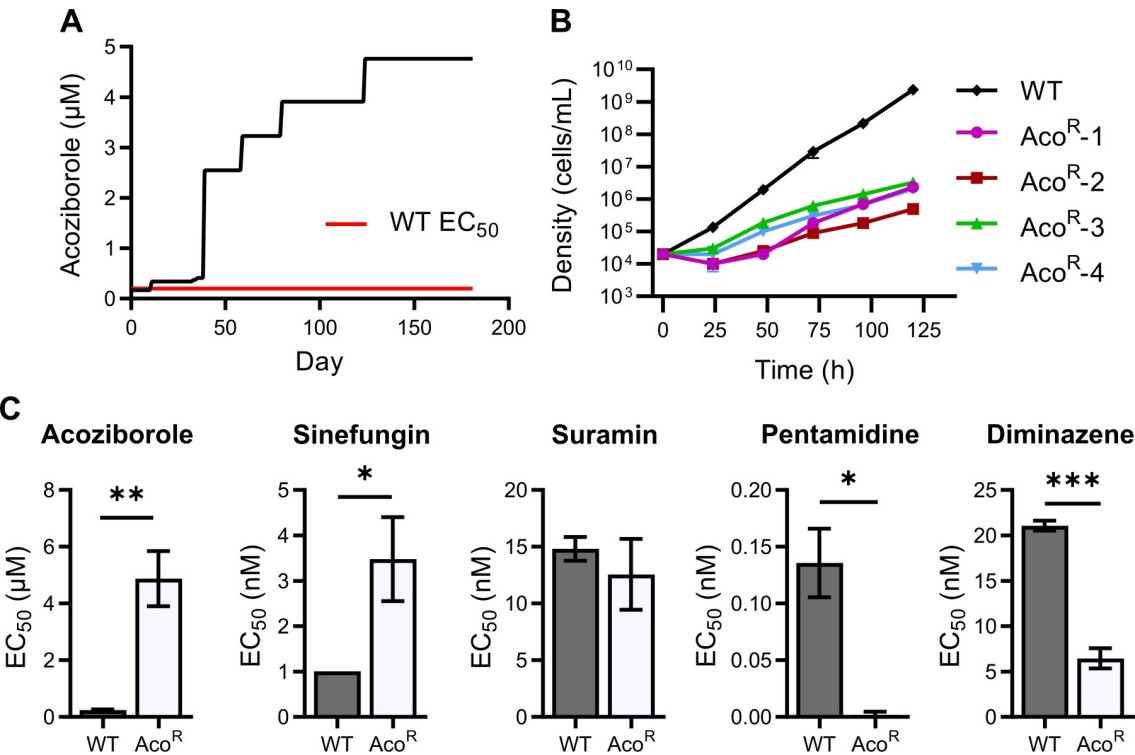

**Fig 1. Generation of an acoziborole-resistant cell line and analysis of cross-resistance.** A) Acoziborole resistance was generated in *T. brucei* Lister 427 cells through the addition of incremental concentrations of the benzoxaborole to cell cultures *in vitro*. Red line indicates mean wild-type EC$_{50}$ for acoziborole. B) Cell density of the four Aco[R] clones, as well as a wild-type control, were monitored daily. Aco[R] cells were grown in the presence of 4.96 μM acoziborole and all cell cultures were passaged (seeding density of $2 \times 10^4$ cells/mL) when density reached $2 \times 10^6$ cells/mL. Cumulative cell density over a period of 5 days (120 hours) is shown (n = 3 per sample group). C) The Aco[R] line showed a significantly increased EC$_{50}$ compared to wild-type cells. Mean EC$_{50}$ was 4.88 ± 0.56 μM, approximately 25-fold higher than WT cells (EC$_{50}$: 0.25 ± 0.01). Interestingly, Aco[R] cells also showed an increased resistance to the AdoMet-dependent MTase inhibitor sinefungin, which was shown to act in similar fashion to acoziborole. Aco[R] cells were hypersensitive to pentamidine and diminazene. All drug sensitivity assays were carried out in duplicate and EC$_{50}$ values were obtained from at least 3 independent experiments.

acozibrole (p = 0.0012), compared to wild-type cells (Fig 1C). Several other compounds were tested to check for cross-resistance or cross-sensitivity (Fig 1C). Aco$^R$ exhibited a 3.5-fold increase in $EC_{50}$ against sinefungin, an adenosine analogue that acts as a methyltransferase inhibitor (Fig 1C). We previously observed antagonism between sinefungin and acoziborole, and sinefungin treatment results in similar metabolic phenotypes to those observed in acoziborole-treated cells [35], suggesting they impact upon similar metabolic processes. In contrast, the Aco$^R$ line exhibited increased sensitivity to both pentamidine and diminazene aceturate (Fig 1C), both of which are thought to act on the trypanosome mitochondrion [24].

To test the stability of the Aco$^R$ line, two clones were further analysed (S1 Fig). Resistant cells were grown in the presence or absence of the benzoxaborole (4.96 μM) for 14 days, after which further drug sensitivity assays were carried out with acoziborole (S1 Fig). Interestingly, whilst still mildly, yet significantly (p = 0.0038 and 0.0031 for clones 1 and 2, respectively) resistant (~3-fold increase in $EC_{50}$), the resistance phenotype appeared to be reversible in Aco$^R$ cells when cultured in the absence of acoziborole (S1 Fig). In addition, Aco$^R$ cells passaged in the absence of drug exhibited an increased growth rate, compared to cells maintained under drug pressure (S1 Fig).

Slow growth of cells is likely to reduce the requirement for RNA processing, which could generate drug resistance in the cases of acoziborole and sinefungin. We therefore examined whether reduction in growth rate, brought about through changes in incubation temperature, could impact on acoziborole sensitivity in wild-type cells (S2 Fig). Cells were maintained in 5% $CO_2$ with lowered temperatures. Whilst a reduction to 34°C did not significantly affect growth rate (doubling times of 7 h and 6.8 h at 37°C and 34°C, respectively), growth was decreased at 30°C (doubling time of 18.5 h). At both temperatures, acoziborole $EC_{50}$ was significantly reduced (fold changes of 1.95 and 3.02 for 37°C vs. 34°C and 37°C vs. 30°C, respectively) compared to 37°C controls (S2B Fig), indicating increased acoziborole sensitivity.

## Aco$^R$ line exhibits a 'stumpy' or 'procyclic' transcriptome

To characterise the Aco$^R$ cell line, and dissect the changes in expression that rendering the parasite drug-resistant, RNA-sequencing was carried out on all four Aco$^R$ clones as well as the parental wild-type Lister 427 and a wild-type passage control cultured in the absence of acoziborole. Reads (mean number of reads: 11,815,635 ± 1,128,622) were aligned to a 'hybrid' genome consisting of 11 core chromosomes from the TREU 927 reference genome complemented by BES contigs from the Lister 427 genome (mean alignment rate: 83.52 ± 0.33%). The resulting alignment files were filtered for primary alignments only (Samtools), before read counts for each gene were calculated (HTseq). Finally, differential expression (DESeq2) as well as polymorphisms (SnpEff) were analysed (see Materials and Methods for details).

The final differential expression dataset consisted of 10,151 genes (after removing genes with read counts of <10). Significant changes in transcript abundance were observed for 4,061 genes based on *Padj* (P-value adjusted by Benjamini-Hochberg correction) alone, although this was reduced to 796 genes when taking into account a Log$_2$ fold change of $\geq 1$ or $\leq -1$ (Fig 2 and S1 Table). Of these, 178 transcripts were significantly increased in abundance, whilst the remaining 618 genes were significantly decreased (Fig 2A and S1 Table).

Strikingly, the majority of genes exhibiting increased transcript abundance (*Padj* $\leq 0.05$, Log$_2$ fold change $\geq 1$) in the Aco$^R$ line have been previously classified as stumpy-form- or PCF-specific genes. In particular, GPEET and EP procyclins (Tb927.6.510, Tb927.10.10260 and Tb927.10.10250 for GPEET, EP1 and EP2, respectively), both surface markers of early and late-differentiated PCFs, respectively, exhibited elevated transcript abundance (Fig 2B and Table 1). A nucleoside transporter, orthologous to TbNT10, previously reported to be

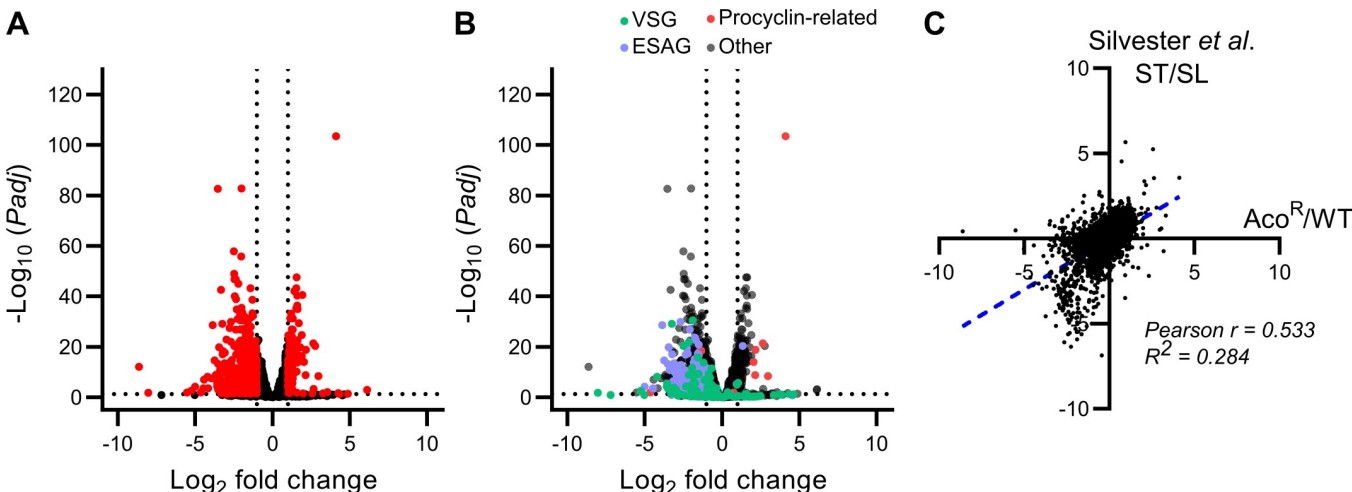

**Fig 2. Overview of transcriptomics analysis of the Aco$^R$ cell line.** A) Volcano plot of statistical significance (*Padj* < 0.05) against fold change (Log$_2$ fold change <-1 or >1) between Aco$^R$ and wild-type cells. Differentiation expression was calculated using DESeq2 [66]. Transcripts undergoing significant differential expression based on the aforementioned parameters are highlighted in red. B) Volcano plot of the same RNAseq data highlighting changes in expression of life-cycle stage-specific surface proteins. Green data points are genes whose production descriptions match "VSG", whilst purple points indicate BSF-associated expression sites (ESAGs). Finally, red data points indicate genes whose annotation include "procyclin", which includes GPEET and EP procyclin. C) Data from this study was compared to previously published transcriptomics data comparing stumpy (ST) form to slender (SL) form *T. brucei* [13]. Raw data from the Silvester *et al.* study [13] was processed by the same means as the data generated here (Aco$^R$ vs wild-type; WT), and log$_2$ fold change (as calculated by DESeq2) of Aco$^R$/WT was compared to ST/SL by linear regression ($R^2$; blue dotted line) and correlation (Pearson's r).

up-regulated in stumpy form parasites, was also elevated on Aco$^R$ cells (Tb927.2.6320; Table 1) [44]. Further examples included pyruvate phosphate dikinase (Tb927.11.6280) and transketolase (Tb927.8.6170), previously shown to be expressed exclusively in PCF *T. brucei* (S1 Table)

**Table 1. Top 20 genes with significantly increased abundance in Aco$^R$ cells.** Only one example shown for repeated genes. For full results see S1 Table.

| Transcript ID | Product Description | Log$_2$ fold change (Aco$^R$/WT) | Padj |
|---|---|---|---|
| Tb927.8.1386 | M5 ribosomal RNA | 6.14 | 6.47E-04 |
| Tb10_snRNA_1:snRNA | uRNA C | 4.88 | 4.40E-02 |
| Tb927.9.1180 | variant surface glycoprotein (VSG, pseudogene), putative | 4.33 | 4.94E-02 |
| Tb927.6.510 | GPEET procyclin | 4.11 | 3.58E-104 |
| Tb927.8.5920 | hypothetical protein | 3.19 | 3.22E-02 |
| Tb927.6.450 | procyclin PARP A | 2.97 | 3.98E-09 |
| Tb927.5.4020 | hypothetical protein | 2.78 | 4.25E-21 |
| Tb927.10.10260 | EP1 procyclin | 2.63 | 4.06E-22 |
| Tb927.10.8340 | hypothetical protein | 2.57 | 3.04E-02 |
| Tb927.10.10250 | EP2 procyclin | 2.16 | 1.36E-19 |
| Tb927.6.480 | surface protein EP3-2 procyclin precursor | 2.14 | 1.61E-09 |
| Tb927.6.520 | EP3-2 procyclin | 2.04 | 9.98E-15 |
| Tb927.10.14060 | hypothetical protein | 1.98 | 2.82E-02 |
| Tb927.11.6280 | pyruvate phosphate dikinase (PPDK) | 1.93 | 2.97E-41 |
| Tb927.9.7470 | purine nucleoside transporter (TbNT10) | 1.90 | 2.10E-25 |
| Tb927.8.6170 | transketolase, putative | 1.89 | 3.49E-17 |
| Tb927.11.2410 | Flabarin, putative | 1.89 | 2.99E-15 |
| Tb11.v5.0854 | NLI interacting factor-like phosphatase, putative | 1.79 | 1.29E-20 |
| Tb927.6.2890 | single strand-specific nuclease, putative | 1.72 | 3.76E-30 |
| Tb927.9.6090 | PTP1-interacting protein, 39 kDa (PIP39) | 1.70 | 1.87E-17 |

[45,46], and a single strand-specific nuclease, reported to be up-regulated in differentiating trypanosomes (Tb927.6.2890) [47]. Finally, abundances of transcripts encoding PAD proteins, as well as PIP-39 (Tb927.9.6090), known to be involved in BSF-to-stumpy differentiation, were also increased in the Aco$^R$ cell line (Tables 1 and S1).

There was also increased abundance in transcripts associated with several transporters, such as AATP11 (Tb927.4.4730; S1 Table), three copies of a pteridine transporter (Tb927.1.2820, Tb927.1.2850 and Tb927.1.28; S1 Table) and three nucleoside transporters (TbNT8.1, TbNT8.2 and TbNT10, Tb927.11.3610, Tb927.11.3620 and Tb927.9.7470, respectively; S1 Table).

Conversely, transcripts specific to BSF-stage parasites exhibited decreased transcript abundance in the Aco$^R$ cell line (Table 2). In particular, there was widespread downregulation of VSGs and expression site associated genes (ESAGs; Fig 2B). Whilst trypanosomatids typically express only one VSG and an associated expression site at any one time, the appearance of numerous VSGs in the transcriptomics dataset are likely the result of short paired-end reads aligning to multiple annotated VSGs that exhibit close homology. Importantly, the predominant VSG in Lister 427, VSG221 (Tb427.BES40.22), was also significantly reduced (Log$_2$ fold change: -1.62, Padj = 0.003; S1 Table). Furthermore, glucose transporters, several adenosine and nucleoside transporters, and glycolytic components such as pyruvate kinase 1 (Tb927.10.14140; S1 Table), a pyruvate transporter (PT1, Tb927.3.4070; S1 Table) hexokinase (Tb927.10.2010 and Tb927.10.2020; S1 Table) and glyceraldehyde 3-phosphate dehydrogenase (Tb927.6.4280 and Tb927.6.4300; S1 Table) were all significantly (p < 0.05) reduced in the acoziborole-resistant cells (S1 Table).

No changes in expression were observed in CPSF3 (Tb927.4.1340; S1 Table), the protein target of acoziborole [33]. There were small but significant decreases observed in mRNA encoding CBP1 (Tb927.10.1030, Tb927.10.1040 and Tb927.10.1050; S1 Table). This gene was

**Table 2. Top 20 genes with significantly decreased abundance in Aco$^R$ cells.** Only one example shown for repeated genes. For full results see S1 Table.

| Transcript ID | Product Description | Log$_2$ fold change (Aco$^R$/WT) | Padj |
|---|---|---|---|
| Tb927.11.16180 | hypothetical protein | -8.62 | 7.92E-13 |
| Tb927.9.16490 | variant surface glycoprotein (VSG), putative | -8.02 | 1.62E-02 |
| Tb427.BES4.9 | expression site-associated gene 11 (ESAG11, pseudogene), putative | -5.01 | 6.62E-05 |
| Tb927.8.1660 | procyclin-associated gene (pseudogene), putative | -4.64 | 1.08E-02 |
| Tb427.BES15.1 | folate transporter, putative | -4.14 | 2.42E-04 |
| Tb427.BES10.1 | transferrin-binding protein, putative | -3.94 | 3.36E-08 |
| Tb927.2.3330 | hypothetical protein | -3.66 | 1.36E-09 |
| Tb927.2.5384 | retrotransposon hot spot protein (RHS, pseudogene), putative | -3.50 | 3.31E-09 |
| Tb927.11.18650 | expression site-associated gene 1 (ESAG1) protein, putative | -3.46 | 1.37E-20 |
| N19B2.120 | RHS5, pseudogene | -3.42 | 1.13E-08 |
| Tb927.5.170 | hypothetical protein, conserved | -3.34 | 7.78E-03 |
| Tb927.2.3340 | hypothetical protein | -3.33 | 2.31E-43 |
| Tb427.BES4.4 | expression site-associated gene 3 (ESAG3, pseudogene), putative | -3.27 | 1.30E-01 |
| Tb927.9.15980 | nucleoside transporter 1, putative | -3.21 | 4.43E-19 |
| Tb927.9.7330 | expression site-associated gene 2 (ESAG2) protein, putative | -3.15 | 4.85E-07 |
| Tb427.BES4.6 | expression site-associated gene 8 (ESAG8) protein, putative | -3.13 | 5.66E-05 |
| Tb927.9.16010 | expression site-associated gene 4 (ESAG4, pseudogene), putative | -3.10 | 2.84E-14 |
| Tb927.6.180 | receptor-type adenylate cyclase GRESAG 4, putative | -2.97 | 1.22E-23 |
| Tb427.BES5.11 | expression site-associated gene 1 (ESAG1) protein, putative | -2.96 | 6.23E-03 |
| Tb927.1.2070 | expression site-associated gene 11 (ESAG11), fragment | -2.83 | 5.28E-03 |

recently shown to be a determinant of resistance to the veterinary benzoxaborole AN11736. However, acoziborole does not possess the valyl-ester linker that serves as a cleavage site for CBP1 [40], and thus, this result is unlikely to be of biological significance.

Sequence polymorphisms were also analysed in the transcriptomics data in order to assess mutations that may be associated with acoziborole resistance. However, none were found associated with genes encoding proteins that are known to be associated with benzoxaborole mode of action or resistance phenotypes in other systems, although the supplementary table (S2 Table) provides the data for future reference as more knowledge about these compounds and their mechanisms come to light.

To test how the transcriptomics data compared to that of BSF-to-stumpy differentiation, further analysis was carried out using previously generated data from two studies [13,48]. Firstly, a study comparing ascending (slender) and peak (stumpy) parasitaemias *in vivo* [13], and secondly a study comparing slender BSFs, stumpy BSFs, as well as early (socially motile) and late (non-socially motile) procyclics [48]. All data were processed as described in the Materials and Methods section. Firstly, $\log_2$ fold changes of stumpy vs. slender (from Silvester *et al.* [13]) were compared to $\log_2$ fold changes of Aco$^R$ vs. wild-type (Fig 2C; S1 Table). Linear regression ($R^2 = 0.284$) and Pearson's correlation coefficient ($r = 0.533$) were calculated, with both showing significant correlation between the datasets ($p < 0.0001$), indicating that Aco$^R$ cells showed a high degree of similarity, albeit at a transcriptome level, to stumpy-form *T. brucei*. Interestingly, a large portion of genes down-regulated in stumpy form cells, but not in acoziborole-resistant cells, were associated with cell division. For example, histones and cytoskeletal proteins (S1 Table).

Furthermore, there was equally strong correlation between the Aco$^R$ vs. wild-type $\log_2$ fold changes when compared to $\log_2$ fold changes of early and late procyclic cells vs. slender cells ($r = 0.505$ and $0.540$ for comparisons to early and late procyclics, respectively; S3 Fig; S1 Table), from Naguleswaran *et al.* [48]. Taken together, these results indicated that the acoziborole resistant line exhibited similarities on a transcriptomic level to both stumpy and procyclic-form parasites.

## Metabolomics analysis of the Aco$^R$ cell line

To further probe changes in the Aco$^R$ cell line as a result of induced resistance to the benzoxaborole, LC-MS was carried out on cell pellets to compare wild-type *T. brucei* to two Aco$^R$ clones, both in the presence and absence (cultured without drug for 48 hours, or 1 passage) of acoziborole resulting in six sample groups (n = 4). A total of 754 putative metabolites were identified in the resulting dataset, of which 75 were matched to an authentic standard (S3 Table). The data was subsequently $Log_2$ transformed and Z-scaled (Log transformation and auto-scaling in Metaboanalyst, respectively), and one-way ANOVA was applied to identify metabolites whose abundance was significantly changed between the sample groups (S3 Table). This resulted in 197 metabolites identified as significantly altered (False discovery rate, FDR < 0.05; Fig 3A, full ANOVA results in S4 Table).

Clustering of the samples using the 197 significantly altered metabolites showed that acoziborole-treated wild-type cells were most divergent from the other 5 sample groups (Fig 3B), indicating that resistant cells both in the presence and absence of acoziborole exhibited a metabolic phenotype closer to that of wild-type cells than that of drug-treated sensitive cells.

The most notable changes involved those in S-adenosyl-L-methionine (AdoMet) metabolism, as previously observed [35] (Fig 3C). In both Aco$^R$ clones, both in the presence and absence of acoziborole, the increases in AdoMet, 5'-methylthioadenosine (5'-MTA) and adenine were absent, suggesting resistant parasites were able to withstand the metabolic impact of

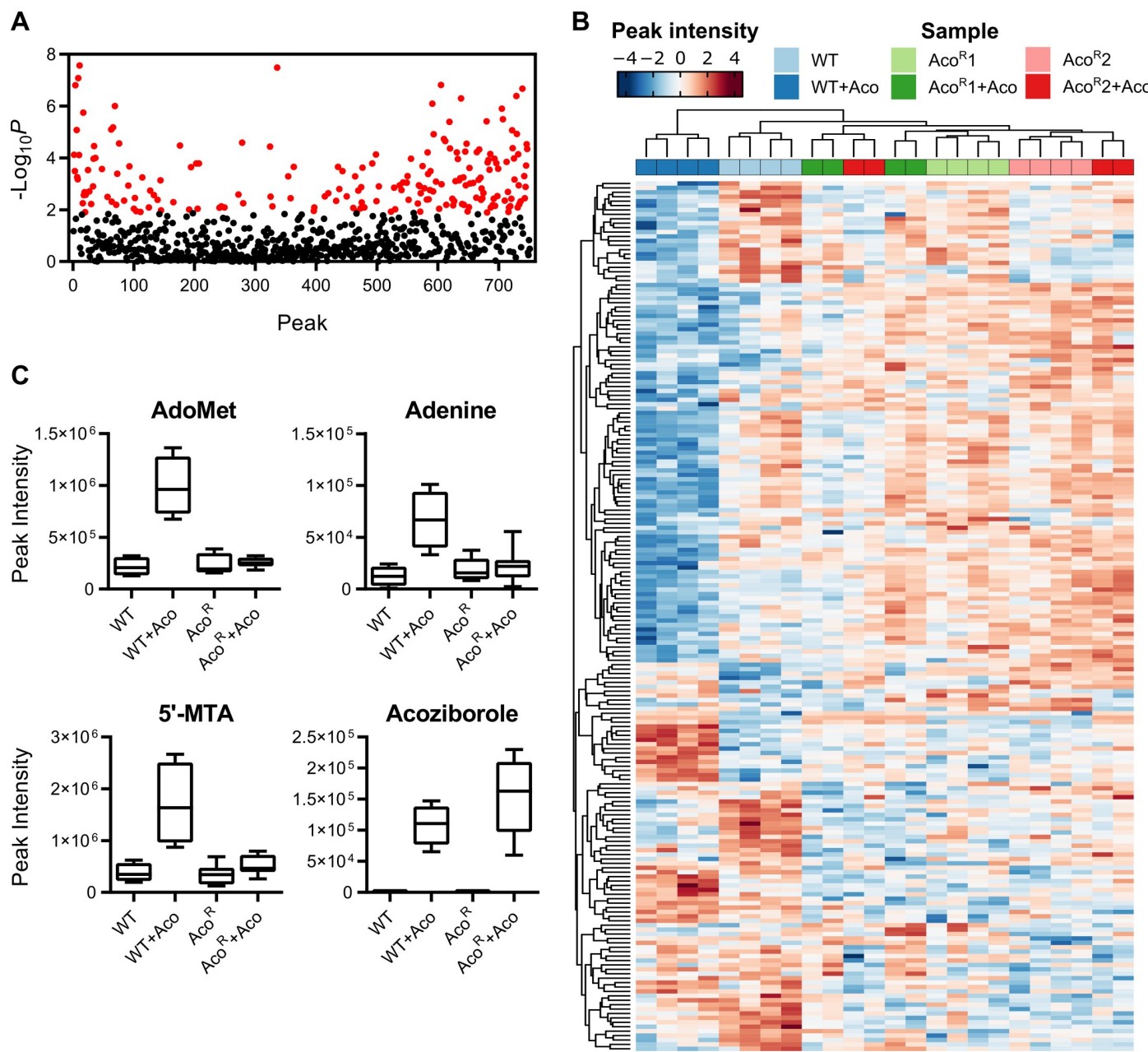

**Fig 3. Metabolomics analysis of Aco[R] cells.** A) Results of an analysis of variance (ANOVA) on 754 putatively identified metabolites. Each peak was assigned a number (S3 Table) for clarity, and significant metabolites (FDR < 0.05; 197 metabolites) are highlighted in red. B) Heat map of significantly altered metabolites (ANOVA, FDR < 0.05). LC-MS peak intensities were log transformed and auto-scaled (mean-centred, divided by the standard deviation for each metabolite). Both samples and metabolites were clustered, showing clear separation of the wild-type, acoziborole-treated samples from all other sample groups. C) AdoMet metabolism is significantly disrupted during acoziborole treatment in wild-type cells, but unchanged in Aco[R] cells irrespective of drug treatment. However, acoziborole is still detected in Aco[R] cell pellets. Abbreviations: Aco, acoziborole; AdoMet, S-adenosyl-L-methionine; 5'-MTA, 5'-methylthioadenosine.

acoziborole treatment. Adenosine levels were elevated in Aco[R] samples (S3 Table), corresponding to elevated transcript levels of adenine phosphoribosyltransferase (Tb927.7.1790; S1 Table). Importantly, intracellular acoziborole abundance (determined through finding a peak at the mass of the drug) remained high in Aco[R] cells, suggesting resistance was not a result of reduced accumulation of the compound across the plasma membrane either through reduced uptake or increased efflux (Fig 3C).

Several other putatively identified metabolites followed a similar trend to AdoMet, including 4-hydroxy-4-methlglutamate, mono-, di- and tri-methyl-L-lysine, 5-guanidino-2-oxopentanoate (2-oxoarginine) and 8-amino-7-oxononanoate (S3 Table), the latter thought to be involved in AdoMet-dependent transaminase reactions leading to biotin synthesis. Taken together, these data suggest that the significant metabolic perturbations that were the hallmark of acoziborole treatment in wild-type *T. brucei* were nullified in resistant cells even though drug uptake appeared to be unaffected. Apart from acoziborole itself, there were no significant differences (*t*-test, FDR < 0.05) between treated and untreated Aco$^R$ cells, highlighting that drug treatment of these resistant cells did not lead to further metabolic perturbations.

Several metabolites were altered only in Aco$^R$ cells compared to WT *T. brucei*. The most significant perturbation restricted to Aco$^R$ cells only was putatively identified, based on its mass and predicted formula, as nonaprenyl-4-hydroxybenzoate (Fig 4A), a metabolite involved in the synthesis of ubiquinone-9 [49], but not previously reported in *T. brucei* and a metabolite for which no authentic standard was available to confirm identity. 2-oxoglutarate, a citric acid cycle intermediate, was also increased in Aco$^R$ cells (Fig 4B). Interestingly, transcripts associated with key enzymes in oxoglutarate metabolism, such as glutamate dehydrogenase (Tb927.9.5900), and the oxoglutarate dehydrogenase complex (Tb927.11.11680, Tb927.11.9980 and Tb927.11.1450; S1 Table), were all elevated in Aco$^R$ cells. Conversely, pyruvate levels were reduced in resistant cells (S3 Table), correlating with reduced pyruvate kinase 1 expression (S1 Table).

Levels of L-carnitine and O-acetylcarnitine (Fig 4C and 4D, respectively) were both elevated (the latter only significantly elevated in untreated Aco$^R$ cells). However, there were no significant transcript changes observed in genes encoding proteins involved in carnitine metabolism, such as carnitine O-acetyltransferase (Tb927.11.2230; S1 Table). One previous study suggested that intracellular carnitine levels in PCF cells are higher than those in BSF cells [50] and therefore, elevated carnitine levels could be the result of increased influx. There were also increases in orotate and (S)-dihydroorotate in the Aco$^R$ line (Fig 4E and 4F, respectively), both precursors of UMP biosynthesis, although a peak corresponding to UMP was not detected, likely due to technical limitations of the mass spectrometer. No transcript changes were detected for genes directly involved in the synthesis of orotate and (S)-dihydroorotate, although significant reductions were exhibited in several genes involved in pyrimidine metabolism (nucleoside diphosphatase, Tb927.7.1930, Log2 fold change = -1.80; uridine phosphorylase, Tb927.8.4430, -1.47; nucleoside hydrolase, Tb11.v5.0270, -1.02; S1 Table). Finally, Aco$^R$ cells had significantly (FDR = 0.0021; S4 Table) reduced levels of L-cystathionine (S3 Table)

Comparison of untreated wild-type cells with untreated Aco$^R$ cells by FDR corrected t-test (FDR < 0.05) showed a total of 52 significantly altered metabolites, the majority of which were identified in the ANOVA analysis outlined above. The statistical outputs are provided in S4 Table.

Metabolomics data were also compared to that of acoziborole-treated PCF cells [35]. Most metabolic changes occurring in wild-type BSF *T. brucei* after acoziborole treatment also occurred in PCF cells, albeit with a significantly higher drug dose, reflective of the higher dose required to kill PCF trypanosomes [35]. However, there were key differences, some of which were also observed in Aco$^R$ cells. For example, a peak putatively identified as 4-hydroxy-4-methylglutamate (a component of C5-branched dibasic acid metabolism) was increased upon acoziborole treatment in BSF *T. brucei*, but reduced in both Aco$^R$ and PCF. PCF trypanosomes also exhibit less significant increases in adenine compared to BSF after acoziborole treatment [35], similar to Aco$^R$ cells. Finally, there was reduced perturbation in other pyrimidines such as cytidine and deoxyuridine in both PCF and Aco$^R$ cells, both of which are reduced in BSF parasites post-treatment (S3 Table). However, because the PCF experiments were carried out independent of this study, it is difficult to directly compare metabolite abundances.

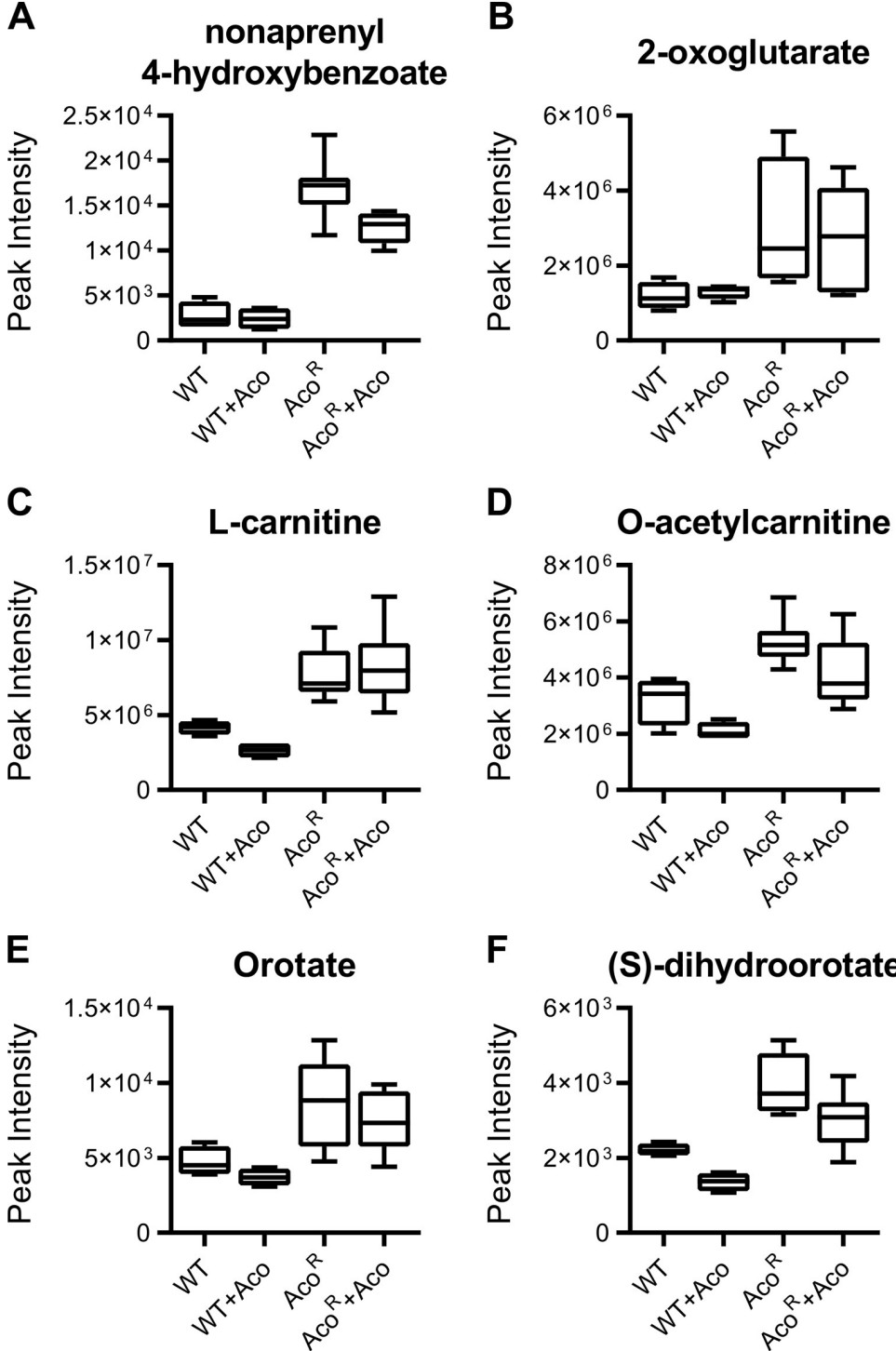

**Fig 4. Metabolites elevated significantly in Aco[R] cells.** Metabolomics data was mined to identify metabolites altered in Aco[R] cells compared to wild-type *T. brucei*. Metabolites identified as nonaprenyl 4-hydroxybenzoate (A) and 2-oxoglutarate (B) support the indications that the Aco[R] is in a differentiated state, as they are involved with ubiquinone and citric acid cycle metabolism, respectively. Levels of both L-carnitine (C) and O-acetylcarnitine (D) are reduced in wild-type cells after acoziborole-treatment. However, both were observed at significantly higher levels in Aco[R] cells both in the presence and absence of acoziborole. Orotate (E) and (S)-dihydroorotate (F) are intermediates of uridine monophosphate and ultimately, uridine triphosphate biosynthesis, which is required for UTP-dependent mitochondrial mRNA polyadenylation. Levels of these metabolites are similarly reduced in acoziborole-treated wild-type cells, but elevated in Aco[R] cells. Abbreviations: Aco, acoziborole.

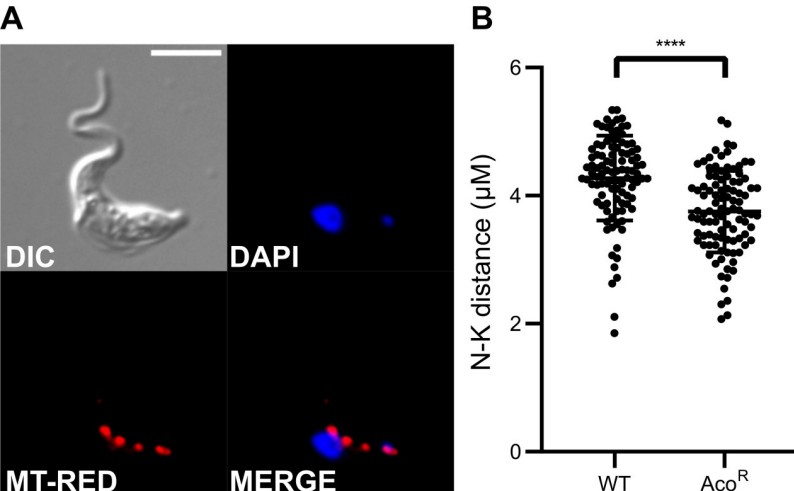

**Fig 5. Microscopy analysis of AcoR cells.** A) Morphology and organelle arrangement in Aco<sup>R</sup> cells was analysed by fluorescence microscopy. Overall morphology appeared to be similar to typical BSF parasites (differential interference contrast; DIC). In addition, kinetoplast DNA was localised to the posterior end of the cell (DAPI). Finally, localisation and shape of the single mitochondrion was observed to be normal (Mitotracker red; MT-RED). Scale bar represents 5 μm. B) The distance between the centre of the nucleus and kinetoplast were measured (n = 100, 3 independent samples) in wild-type and Aco<sup>R</sup> cells. Results were analysed for significance by applying an unpaired *t*-test (****p $< 0.0001$).

## Aco<sup>R</sup> cells retain BSF morphology

The morphology of *T. brucei* differs significantly depending on life cycle stage. BSFs are long and slender, whilst stumpy-forms are rounded and short, as their name implies. In contrast, PCFs are elongated and exhibit decreased distance between kinetoplast (mitochondrial DNA) and nucleus [51].

Given the "stumpy"- or "procyclic"-like profile observed on both transcriptome and metabolome levels, we analysed Aco<sup>R</sup> cell morphology (Fig 5), to determine whether the PCF-like phenotype was also reflected in cell shape and size, as well as organelle arrangement. Overall morphology appeared to resemble that of classical BSF *T. brucei* (Fig 5A). In addition, mitochondrial arrangement was no different to that of wild-type BSF *T. brucei* (Fig 5A). In this life cycle stage, the mitochondrion remains metabolically inactive, and is present as a single tube stretching throughout the cell. Although the localisation of the nucleus and kinetoplast, the organelle containing mitochondrial DNA, appeared to be normal in Aco<sup>R</sup> lines, the distance between the two was further analysed in both wild-type and Aco<sup>R</sup> cells (Fig 5B). A small, but significant (Student's *T*-test, $P < 0.0001$) difference in nucleus-kinetoplast distance was present, suggesting that the nucleus and kinetoplast localise closer together in the Aco<sup>R</sup> cell line.

## Discussion

Subspecies of the protozoan parasite *Trypanosoma brucei* are the causative agents of HAT, a debilitating disease prevalent across the sub-Saharan African continent. Whilst case numbers continue to decline, there is an urgent need for safe, effective and orally bioavailable chemotherapeutics if the target of elimination is to remain feasible.

Acoziborole is a novel trypanocide currently in clinical trials and has shown great potential as a front line drug to combat both early- and late-stage HAT caused by both *T. b. gambiense* and *T. b. rhodesiense*. The drug is effective in a single dosing, representing a truly remarkable

step change in possible therapy of HAT. While a mode of action of acoziborole and a related veterinary benzoxaborole, and their protein target, CPSF3, were recently identified [32,33], less is known about potential parasite mechanisms of resistance against the compound. One study, however, did reveal that one of several genomic changes was an increase in copy number of the *CPSF3* gene, pointing to a potential mechanism where increased target concentration causes a proportional resistance to the drug [39]. We attempted to ascertain whether expression or sequence of CPSF3 (Tb927.4.1340), varied compared to wild-type. However, no altered expression nor sequence was observed in this gene in our resistant line. Unfortunately, in the absence of data other than transcript abundance we cannot definitively exclude a role for CPSF3.

This study outlines the characterisation of a laboratory-derived acoziborole-resistant *T. brucei* cell line using omics-based technologies. The Aco$^R$ line exhibited high levels of acoziborole resistance and interestingly, cross-resistance to sinefungin, an AdoMet-dependent methyltransferase inhibitor. We previously found similarities between the metabolic phenotypes of acoziborole- and sinefungin-treated cells [35]. In addition, when used simultaneously, the interaction appears to be antagonistic, suggesting inhibition of similar biological processes, in this case, mRNA maturation. Aco$^R$ cells also exhibited hypersensitisation against pentamidine and diminazene. Uptake of the latter is known to be mediated by the TbAT1 transporter (or P2; Tb927.5.286b), although transcripts associated with this gene were down-regulated in the resistant cells (S1 Table). Pentamidine also enters via TbAT1, but there is also major uptake via the Aquaglyceroporin 2 (AQP2) transporter, although transcript abundance of this gene (Tb927.10.14170) remained unchanged in acoziborole resistant cells. Thus, hypersensitisation to diamidines would appear to be unrelated to transcript levels of the genes encoding the key transporters of these compounds.

Remarkably, transcriptomics analysis of the resistant line suggests that the parasites underwent a "partial differentiation event" in acquiring acoziborole resistance. This is emphasized by increased abundance of transcripts associated with stumpy and procyclic forms, such as EP/GPEET procyclins and PTP-1, as well as upregulation of several PCF-specific metabolic genes such as transketolase and pyruvate phosphate dikinase, the latter not expressed in BSF *T. brucei* [45,46,52]. Transcripts associated with proteins involved in stage-specific regulation of mRNA (e.g. increases in PAD1, PAD2, ZC3H11 and ZC3H22; Tb927.7.5930, Tb927.7.5940, Tb927.5.810 and Tb927.7.2680, respectively) were also significantly altered. Concomitantly, Aco$^R$ cells exhibited reduced abundance of BSF-stage transcripts such as VSG and the THT-1 glucose transporters. Given the apparent loss of variant surface glycoproteins and ESAGs, known to be essential to successful growth in mammalian blood [53,54], we did not test the ability of these cells to survive in mammals where we would presume them to be unviable.

A significant limitation in this study is the lack of protein level data to complement the transcriptional data outlined here. We have assumed that transcription acts as a proxy for the cellular proteome, but additional work, such as proteomics or transmission electron microscopy is required to verify this.

Interestingly, in the context of the data outlined here, Jones and colleagues found genomic deletions in one resistant cell line in a region corresponding to the AdoMet decarboxylase (AdoMetDC) array [39], but we could not identify similar alterations in the Aco$^R$ line. There was, however, a small but significant reduction in transcript abundance of a tRNA-methyltransferase (Tb927.6.4420, S1 Table). This gene underwent a deletion in acoziborole-resistant cells generated in the aforementioned study [39].

Recent studies in *T. brucei* have highlighted similarities between the pathways controlling differentiation and those controlling stress responses [55,56]. Furthermore, treatment with suramin, a frontline drug for the management HAT, was recently shown to lead to an increase

in mitochondrial ATP production and expression of proteins normally associated with PCF trypanosome metabolism [57]. Further work must be carried out, including on a proteomic level, to determine whether the stumpy/PCF phenotype resulting from generation of resistance to acoziborole is a direct response to benzoxaborole action, or a conserved global response to drug pressure or other pressures. Our data indicate that removing drug pressure reverses the resistance phenotype significantly, and this is coupled to increased growth rate. However, our data suggests that induction of stress by reduction of temperature, which is also coupled to reduced growth rate, is not sufficient to generate acoziborole resistance and thus, there remain unidentified mechanisms involved in generating resistance to this compound.

Notably, this study used the Lister 427 *T. brucei* cell line, which is known to be monomorphic and carries defects in the classical trypanosome differentiation pathway [42]. Drug resistance in African trypanosomes typically arises through the loss of drug-uptake transporters [22,58], increased drug efflux [59], or overexpression of the drug target [33], and the resistance phenotype uncovered in this study was unexpected. Given these data, it would be worthwhile to further investigate acoziborole resistance in pleiomorphic *T. brucei* to test whether the acoziborole resistance mechanism described here can also occur in normally differentiation competent cells.

Although we have not identified the specific mechanism of resistance, the work we report here has demonstrated that a global switch in transcriptome from a typical BSF type to a stumpy/PCF-like type can accompany selection of resistance in *T. brucei*. Procyclic forms are less susceptible to acoziborole than BSF ($EC_{50}$: ~0.2 μM and ~1.5 μM for BSF and PCF, respectively; [35]), hence some distinguishing aspect of PCF physiology might underlie the change in sensitivity and a change of transcription profile could be sufficient to select for resistance. The fact that the CPSF3 target is necessary for both BSF and PCF parasites could indicate the presence of other, as yet unidentified targets for acoziborole in *T. brucei*, pointing to a degree of polypharmacology associated with benzoxaboroles. The mechanism of global transcriptional change is unlikely to be relevant to selection of resistance to these drugs in clinical practice as the parasites require a VSG coat that would appear to be lost (albeit, at the transcriptome level) in the process and also have biochemical adaptations essential to living in the bloodstream that differ from those required for PCF to live in the tsetse fly midgut.

## Materials and methods

### *T. brucei in vitro* culture and generation of acoziborole resistance

Bloodstream form (BSF) *T. brucei* culture was carried out using the 427 Lister strain [43]. Cells were cultured in HMI-11 (supplemented with 10% foetal bovine serum) and maintained at 37°C, 5% $CO_2$. Cultures were maintained at densities of $2 \times 10^4$–$2 \times 10^6$ cells/mL. *In vitro* resistance to acoziborole was initiated by maintaining cells in 170 nM of the compound, which was deemed the highest concentration wild-type cells could tolerate long term. Once a cell line was established in this concentration, cells were then transferred to 6 wells of a 24-well plate at a density of $1 \times 10^5$ cells/mL. Six incrementing concentrations of acoziborole were then added, to obtain six 2 mL cultures. Typically, increments of 0.2 μM were used, and cells were incubated at 37°C and 5% $CO_2$. Cells were routinely observed by microscopy and after >2 weeks, live confluent cells in the highest concentration of acoziborole were cultured for 2 passages in the same concentration of the benzoxaborole, before transfer to 6 new wells in a 24-well plate, from which point the process was repeated using the new acoziborole concentration as a starting point. A wild-type cell line was grown in the absence of drug as a "highly-passaged" control, in order to detect transcriptome changes related to *in vitro* culture adaptation. Once

sufficient resistance was deemed to have been generated, cells were cloned by dilution and 4 isolates were recovered.

For quantitative growth analysis, cultures were maintained at the densities described above. Growth curves were plotted cumulatively, by taking the dilution factor after passage into account. Growth rates were calculated using the exponential (Malthusian) growth model algorithm in GraphPad Prism (v8.4.0; www.graphpad.com).

## Alamar blue assays

To obtain *in vitro* $EC_{50}$ values for specific compounds targeting *T. brucei*, alamar blue assays were employed (adapted from [60]). This colorimetric assay was carried out in solid white flat-bottomed 96-well plates. Compounds were added starting with the highest concentration (typically 100 μM) and serially diluted 1:2 over 23 wells, leaving one negative control. Subsequently, cells were added at a final density of $2 \times 10^4$ cells/mL. Plates were incubated for 48 hours at 37°C, 5% $CO_2$, after which 20 μL of alamar blue reagent (resazurin sodium salt, 0.49 mM in 1× PBS, pH 7.4) was added to each well, and the plate incubated for a further 24 hours. In experiments involving the $Aco^R$ cell line, the incubation time after addition of alamar blue was 48 hours for all sample groups.

Reduction of the alamar blue reagent was measured as a function of cell viability on a BMG FLUOstar OPTIMA microplate reader (BMG Labtech GmbH, Germany) with $\lambda_{excitation}$ = 544 nm and $\lambda_{emission}$ = 590 nm. The raw values were plotted against the log value of each concentration of drug or compound (M), and $EC_{50}$ values were calculated using a non-linear sigmoidal dose-response curve. Each assay was performed in duplicate, and $EC_{50}$ values represent a mean of three independent experiments.

## Transcriptomics & data analysis

Total RNA was extracted from in vitro cultures of $10^8$ cells. RNA was purified using a commercial kit (Nucleospin RNA, Macherey-Nagel). A DNase treatment step was included in the kit protocol. RNAseq was carried out by Glasgow Polyomics. The RNA library was prepared using PolyA selection using the TruSeq stranded mRNA sample prep kit (Illumina) and sequencing was carried out using Illumina NextSeq500 sequencing apparatus.

Sequencing files were processed as follows: Raw reads were trimmed and read quality and coverage was assessed using FastQC [61]. Once all sequencing data was judged to be of good quality, the reads were aligned to a hybrid genome consisting of TREU 927 (TriTrypDB; v50.0) core chromosomes complemented with Lister 427 (TriTrypDB; v50.0) BES contigs using HiSat2 (parameters:—no-spliced-alignment) [62,63]. Further filtering and removal of duplicates was done using Samtools (parameters for samtools view: -bS–q 1; parameters for samtools sort: -O BAM) [64]. To analyse differential expression, reads were first counted using htseq-count (parameters: -s reverse -f bam -t exon -i Parent -m union—non-unique-all), part of the HTSeq python library [65], before analysis of expression with DESeq2 [66]. For SNP and indel analysis, raw data was aligned to the Lister 427 genome (427 2018, v50.0) as described above, and data was further processed using the genome analysis tool-kit (GATK) [67]. SNPs and indels were filtered using the SnpEff/SnpSift package [68]. RNAseq data is available at GEO (Accession number: GSE168394).

## Metabolomics, LC-MS & data analysis

Samples for metabolomics analysis were acquired by rapidly quenching $8 \times 10^7$ cells in log phase in a dry-ice/ethanol bath, to ~4°C. For each sample group, four replicates were grown independently. After quenching, samples were centrifuged for 10 minutes at $1,500 \times g$, 4°C,

and all experimental steps hereafter were carried out at 4˚C. Supernatant was poured off and cells resuspended in the remaining supernatant. Samples were then transferred to a 1.5 mL eppendorf tube prior to another centrifugation step at $1,500 \times g$ for 5 minutes. Remaining supernatant was carefully removed, and the cells resuspended in 200 μL extraction solvent. All samples, including a blank and fresh medium control, were then left on a shaker at 4˚C for one hour. Subsequently, samples were centrifuged at $16,060 \times g$ for 10 minutes, and the supernatant was transferred to a 2 mL screw-top tube. A quality control sample was generated by pooling together 10 μL from each sample. Finally, air was displaced with argon gas, and samples were stored at -80˚C until they were analysed by liquid chromatograph-mass spectrometry.

Liquid chromatography-mass spectrometry was carried out by Glasgow Polyomics. Metabolomics samples were separated by HPLC using a ZIC-pHILIC (polymer-based hydrophilic interaction liquid chromatography) column (Merck). Two solvents were used in the column. Solvent A was 20 mM ammonium carbonate in $H_2O$ and solvent B was 100% acetonitrile. Mass detection was carried out using an Exactive Orbitrap mass spectrometer (Thermo). The mass spectrometer was run in positive and negative mode with an injection volume of 10 μL and a flow rate of 100 μL/minute.

Raw mass spectrometry data was converted to mzXML format and split into positive and negative polarity using msconvert [69]. Files were then converted to peakML files with XCMS, which were further processed using mzMatch [70] and Ideom [71]. Data analysis was carried out using Ideom and Metaboanalyst [72]. Data is available at Metabolights (Accession number: MTBLS2559).

### Preparation of slides & microscopy

Mitochondria were stained using Mitotracker Red (Invitrogen) prior to fixation and mounting. Cells (1 mL at a density of $5 \times 10^5$ cells/mL) were incubated for 5 minutes at 37˚C, 5% $CO_2$, with a final concentration of 100 nM Mitotracker. Cells were subsequently centrifuged for 5 minutes at $1,500 \times g$ and washed twice in fresh medium before fixation by addition of a final concentration of 2% formaldehyde in PBS, and a 15-minute incubation at room temperature. Samples were then washed with PBS and transferred onto a poly-L-lysine-coated slide, which was left to air-dry in a biological safety cabinet. Dried slides were rehydrated and washed in PBS, and a counterstain consisting of 1× PBS with 3 μM 4,6-diamidino-2-phenylindole (DAPI) was applied to the slide, before mounting with a coverslip that was sealed with clear nail varnish. Slides were analysed with a Zeiss axioscope (Scope.A1, Zeiss).

To measure the distance between nucleus and kinetoplast, images were obtained from DAPI stained samples and these imported into the Fiji software [73]. Distances were measured after the scale was set using the "measure" tool. For each sample group, 100 measurements were taken in total from three independent microscopy experiments (30–40 measurements per sample group).

### Computational methods

Graphical representation of data was generated using the Graphpad Prism software (v8.4.0; www.graphpad.com) or R [74]. Statistical analyses were carried out using Graphpad Prism, Microsoft Excel or R.

### Supporting information

**S1 Fig. Analysis of reversibility of acoziborole-resistance.** Two clones of the Aco[R] line were grown for 14 days in the presence (+Aco) or absence (-Aco) of 4.96 μM acoziborole, prior to testing sensitivity to the benzoxaborole, compared to a wild-type (WT) control. A) Sigmoidal

dose-response curves of two clones in the presence or absence of acoziborole with a wild-type control. A shift to the right indicates increased acoziborole resistance. B) Mean $EC_{50}$s from three independent experiments. Acoziborole resistance is reversed in $Aco^R$ cells grown without drug pressure, although resistance is still significant compared to that of wild-type cells (Student's T-test, **$p < 0.01$, ****$p < 0.0001$). C) Cumulative growth curves of wild-type *T. brucei* and $Aco^R$ cells in the presence or absence of 4.96 μM acoziborole. Mean doubling times were 7.6 h, 10.8 h, 12.7 h, 7.1 h and 6.8 h for WT, clone 1 +Aco, clone 2 +Aco, clone 1 -Aco and clone 2 -Aco, respectively.
(TIFF)

**S2 Fig. Acoziborole sensitivity under temperature-induced slow growth conditions.** Wild-type *T. brucei* was grown at lower temperature (34˚C and 30˚C) to test the effect of reduced growth rate on acoziborole sensitivity. A) Growth was significantly reduced at 30˚C (doubling times: 7.0 h, 6.8 h and 18.5 h at 37˚C, 34˚C and 30˚C, respectively, calculated using an Malthusian growth model) only. B) Acoziborole sensitivity was significantly increased in *T. brucei* when cultured under conditions of lower temperature and reduced growth rate. Statistics performed by unpaired t-test, **$p < 0.01$.
(TIFF)

**S3 Fig. Comparative RNAseq analysis of $Aco^R$ cells with stumpy and procyclic cells.** Transcriptomics data generated by Naguleswaran and colleagues [48] was processed by the same means as the data generated in this study and $\log_2$ fold changes (as calculated by DESeq2) of $Aco^R$ vs. WT were compared to $\log_2$ fold changes of stumpy vs. slender (A), early procyclic vs. slender (B) and late procyclic vs. slender (C). In this study, early procyclics showed coordinated social motility whilst late procyclics did not [48]. Significance of correlations between the datasets were tested by linear regression ($R^2$; blue dotted line) and Pearson correlation (Pearson's r). Abbreviations: St: stumpy; Sl: slender; Ea: early procyclic; La: late procyclic.
(TIFF)

**S1 Table. Excel file containing differential gene expression analysis comparing the acoziborole-resistant cell line to wild-type *T. brucei* as output by DESeq2.** The dataset is divided into 4 worksheets. The first contains DESeq2 output from the $Aco^R$ cell line analysis; the second contains HTSeq-count output for each sample used in this study. The final two worksheets contain the comparisons of the DESeq2 output from this study to previously published comparisons of slender BSF vs. stumpy form [13], and slender BSF vs. PCFs [48]. These worksheets also contain columns with calculated distance from an "X = Y" line for each gene, in both comparisons. Hypothetically, if $\log_2$ fold change for a gene in the $Aco^R$/WT comparison was equal to that from the other comparisons, the gene would fall on an X = Y line when plotted on a scatter plot. These columns are the calculated deviation from this line for each gene. Positive values indicate a higher $\log_2$ fold change in the $Aco^R$/WT dataset, and conversely, negative values indicate a lower $\log_2$ fold change in the $Aco^R$/WT dataset, when compared to the aforementioned data.
(XLSX)

**S2 Table. Excel file containing SNP and indel analysis of the $Aco^R$ cell line.** The dataset was generated using the SnpEff tool.
(XLSX)

**S3 Table. Excel file containing results of metabolomics analysis of the $Aco^R$ cell line (Ideom output).**
(XLSX)

**S4 Table. Excel file containing outputs of the statistical analyses of the metabolomics data.** (XLSX)

## Acknowledgments

The authors would like to thank Glasgow Polyomics for technical assistance with metabolomics and transcriptomics experiments and Glasgow Imaging Facility for technical assistance with microscopy.

## Author Contributions

**Conceptualization:** Pieter C. Steketee, Isabel M. Vincent, Kathryn Crouch, Fiona Achcar, Nicholas J. Dickens, Annette MacLeod, Michael P. Barrett.

**Data curation:** Pieter C. Steketee, Federica Giordani.

**Formal analysis:** Pieter C. Steketee, Nicholas J. Dickens, Michael P. Barrett.

**Funding acquisition:** Annette MacLeod, Michael P. Barrett.

**Investigation:** Pieter C. Steketee, Federica Giordani, Isabel M. Vincent, Fiona Achcar, Annette MacLeod, Michael P. Barrett.

**Methodology:** Pieter C. Steketee, Federica Giordani, Kathryn Crouch, Fiona Achcar, Nicholas J. Dickens, Annette MacLeod, Michael P. Barrett.

**Project administration:** Michael P. Barrett.

**Resources:** Michael P. Barrett.

**Supervision:** Nicholas J. Dickens, Annette MacLeod, Michael P. Barrett.

**Validation:** Pieter C. Steketee, Isabel M. Vincent, Fiona Achcar, Michael P. Barrett.

**Visualization:** Pieter C. Steketee.

**Writing – original draft:** Pieter C. Steketee, Michael P. Barrett.

**Writing – review & editing:** Pieter C. Steketee, Liam J. Morrison, Michael P. Barrett.

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
