## [Decision Letter · Decision Letter 0]

23 Apr 2021

Dear Prof. Barrett,

Thank you very much for submitting your manuscript "Transcriptional differentiation of Trypanosoma brucei during in vitro acquisition of resistance to acoziborole" for consideration at PLOS Neglected Tropical Diseases. As with all papers reviewed by the journal, your manuscript was reviewed by members of the editorial board and by several independent reviewers. 

All three feel the work is important and of interest to the field but they raise a number of important issues that need to be addressed in a revised manuscript before we can make a decision about acceptance.

In light of the reviews (below this email), we would like to invite the resubmission of a significantly-revised version that takes into account the reviewers' comments. Your revised manuscript is also likely to be sent to reviewers for further evaluation.

Sincerely,

Margaret A Phillips, Ph.D.

Deputy Editor

Margaret Phillips

Deputy Editor

Your manuscript has been reviewed by three experts in the field. All three feel the work is important and of interest to the field but they raise a number of important issues that need to be addressed in a revised manuscript before we can make a decision about acceptance.

Reviewer's Responses to Questions

**Key Review Criteria Required for Acceptance?**

**Methods**

-Are the objectives of the study clearly articulated with a clear testable hypothesis stated?

-Is the study design appropriate to address the stated objectives?

-Is the population clearly described and appropriate for the hypothesis being tested?

-Is the sample size sufficient to ensure adequate power to address the hypothesis being tested?

-Were correct statistical analysis used to support conclusions?

-Are there concerns about ethical or regulatory requirements being met?

Reviewer #1: The methods are appropriate but more extensive analysis is needed for the transcriptome and genome, using the well-annotated TREU927 genome as a basis.

Reviewer #2: Yes to all questions asked except for the question about ethical/regulatory concerns where the answer is no. In the summary and general comments my concerns are raised - most can be addressed by the authors without further experiments but there are straightforward immunoblotting or classic TEM analysis for the authors to consider and which would lift the work.

Reviewer #3: The objective of the study was to examine possible mechanism of acoziborole resistnace in Trypanosoma brucei through the in vitro generation of resistant lines. The study design is appropriate and the trypanosome strain chosen is the one that is mosyt often used for these kinds of in vitro analyses. There is one issue with the trypanosome strain used and that is that the Lister 427 line is monomorphic and has some defects in the normal trypanosome differentiation pathway, and since the mechanism of resistance uncovered in this study appears to involve a differentiation-type switch in gene expression it would have been useful to use a pleiomorphic line as well. However the existence of this mechanism couldn't have been predicted at the start of the study and so the strain used was appropriate given the more likely mechanisms of resistance (these being drug efflux or overexression of the protein target). It may be the monomorphic nature of the strain used that ensured there was no morphological change to accompany the transcriptomic changes.

Sample sizes and statistics are fine.

There are no ethical or regulatory issues involved.

**Results**

-Does the analysis presented match the analysis plan?

-Are the results clearly and completely presented?

-Are the figures (Tables, Images) of sufficient quality for clarity?

Reviewer #1: Benzoxaboroles are likely to be decisive tools in the future control of African trypanosome diseases. The primary target of the drugs that are currently in late-stage trials is the cleavage and polyadenylation factor CPSF3, but various aspects of the drug effects remain unexplained- notably, changes in methylated metabolites. Resistance is always a potential problem for chemotherapy, but for the human candidate, acoziborole, no lines with more than low-level resistance have been described. In this paper, the authors describe T. brucei that are roughly 20-fold resistant to acoziborole. They show changes in methylated metabolites, with no further effects upon treatment. Although the mechanism is-as before, completely unknown, the results do hint at some polypharmacology which may contribute to the severe difficulty in obtaining resistance. The resistant cells also grow very slowly, which would almost certainly preclude selection in the field. However the slow growth could also be the reason why the cells tolerate the drug.

The authors claim that the cells are procyclic-like in their transcriptomes but actually they don't know because they have not done a quantitative comparison with procyclic- or stumpy-form transcriptomes. From the growth kinetics, morphology and transcriptome results, I suspect that these cells are actually showing some characteristics of stumpy-form trypanosomes, more than procyclics.

Major modifications required: The sequencing alignment was done with the most recent genome of the cognate strain, Lister427. While this is essential for examining VSG expression sites, it is pretty useless for everything else. This is because the annotation of that genome is really poor. It is impossible to tell, from these Tables, what gene products are actually affected. More importantly, it is not possible compare the transcriptome results presented here with any of those previously published. It is therefore essential to supply equivalent tables made using the TREU927 genome annotation, which is way superior for everything except expression sites. From my own experience, I can say that it will be essential to re-do the sequence alignments, because the Tables of homologues that can be downloaded from TritrypDB are incomplete. One table (or sheet) should include raw read counts so that it is possible for others to do quantitative comparisons without repeating the alignments. Please include both the open reading frames and the 3'-UTRs; this will enable genes with similar ORFs, but different regulation (such as PGKB and PGKC) to be distinguished. The authors must also then do a comparison with published stumpy-form and procyclic-form transcriptomes, to see which most closely resembles the resistant cells; and also look at cell-cycle regulation. It is indeed revealing that in the discussion, the authors revert to the 927 numbers - referring the reader to supplementary Tables on which these numbers cannot be found! In fact in Table S1 the authors do not even show the 427 gene IDs of the increased and decreased transcripts.

Exactly the same applies to the SNP and indel analysis. You can't tell if something is important if you don't know the gene function. The whole analysis also needs to be done again using 927. You can check afterwards whether the changes are simply differences between 427 and 927.

Table 1: Please compress this to include only one example of the repeated genes. Simplify the annotations (not just copy-paste of the database entry) and show GeneDB numbers for the homologues (if present) in TREU927. The RHS proteins are known to have various different functions: which ones are affected here? Which nucleoside transporter is it, most are now characterised? For example, Tb427_000257300.1 turns out to most likely be an ESAG3 pseudogene, but I had to blast the nucleotide sequence onto 927 to find out. What are all the other "hypothetical proteins" in the Table? Aligning to 927 may well tell you. The results would also appear to suggest a decrease in expression site transcription. Please show a screen shot of the read alignment across the active expression site, then it will be possible to tell.

Tables in text: Put all Padj values in the same format. Tables everywhere: These results are not accurate to 9 significant figures, please remove superfluous decimal places, it makes the Tables much smaller and much easier to read.

Reviewer #2: Yes, but some revision of Fig 1 is needed and revision of Table 2 highly encouraged - see 'Summary and General Comments'

Reviewer #3: Most of the results are clearly presented, there are a couple of minor issues: 

Lines 424-426 Fig 1. Legend to part B does not refer to the figure but talks about changes in EC50 whereas the figure is about doubling times. Also the doubling times should be given quantitatively as numbers rather than just showing growth curves (Lines 122-123)

Line 426 The C is missing where the legend should refer to part C.

Fig 5A - surely tha authors could have got a better image of a single cell rather than one that is folded back on itself.

**Conclusions**

-Are the conclusions supported by the data presented?

-Are the limitations of analysis clearly described?

-Do the authors discuss how these data can be helpful to advance our understanding of the topic under study?

-Is public health relevance addressed?

Reviewer #1: It seems very likely that the cells are acoziborole resistant because they are growing very slowly. Slow-growing parasites have a lower requirement for RNA processing than normal cells. This could also explain the resistance to sinefungin. If this is true then the question is, WHY are these cells growing slowly? Some hints may come from the reanalysis of the transcriptome and genome proposed above.

The conclusion that there is a "master-switch" controlling differentiation is not supported buy the data and should deleted. Instead, I suspect that the changes seen in these cells may be related to a prolonged stress response, as suggested in a recent publication (10.1016/j.pt.2020.11.003). The cited publication also includes numerous references to stresses that can cause stumpy-like changes. Please read, consider, and cite. The question would then be why the cells retain this phenotype. Perhaps the resistant cells have mutated so as to make the stressed state permanent? could this explain the increased susceptibility to some other drugs? Is the phenotype lost really rapidly when selection is removed, or do the authors have to wait to select revertants? If the latter, the change must be genetic and the authors might be able to find it from the genotype results - especially if they also sequence revertants.

Reviewer #2: Overall yes, but clarity in some areas is called for in 'Summary and General Comments'.

Public health relevance is addressed nicely.

Reviewer #3: The major problem with the conclusions is that all of the gene expression data is based on RNA levels and it is known that trypanosomes employ translational control as well so increased transcript numbers may not result automatically in increased protein levels. It would have been useful if the authors had demonstrated that some of the procyclic specific genes were actually expressed at the protein level (for example the procyclins/PARP/ EP/GPEET) as there are antibodies available and it would be easy to demonstrate surface expression of these proteins. Or if they had employed ribosomal profiling rather tha just RNA-seq to demonstrate that the upregulated PCF genes were being translated. In this context one experiment that should be done is to check the protein expression level of CPSF3 since they dismiss the idea of overexpression by saying there were no changes in RNA level but if the CPSF3 mRNA is more efficiently translated in a "PCF-like" context then they may be missing the actual resistance mechanism.

With the metabolomics data they disucss several metabolites with changes in the resistant cell lines eg carnitine, orotate etc but they do not mention whther these metabolic changes link to any upregulated or downregulated genes in the RNA-seq data. This should be done and it should be made clear if there are any significant chnages in pyrimidine metabolic genes for example or genes related to L-carnitine metabolism or pathways L-carnitine is involved in.

I cannot see anywhere where the authors have tested the stability of this resistance phenotype - is the phenotype lost in the absence of acoziborole, and if so how rapidly does it revert? In this light it would also have been interesting to see whether any live trypanosomes could have been recovered from an infected mouse (Lines 290-291).

**Editorial and Data Presentation Modifications?**

Reviewer #1: Minor corrections (grammar or unclear):

"There were small but significant decreases observed in three copies of the CBP1 gene". This should be changed to "There were small but significant decreases observed in mRNA encoding CBP1" followed by an reminder of what the function of CBP1 is, with an appropriate reference.

"HAT case number has" -> "HAT case numbers have" OR "The HAT case number has"

for stage 1 and stage 2 treatment of HAT -> for treatment of stage 1 and stage 2 HAT

downregulated, -> down-regulated; upregulated -> up-regulated.

"T. brucei brucei is infectious to livestock" -> T. brucei brucei and T. brucei rhodesiense are infectious..

the disease can be fatal.. -> the disease is usually fatal.. (I think this is still true)

procyclic (PCF) insect vector stages in the tsetse fly -> delete "insect vector"

"reduction in case number of the disease caused by T. b. gambiense" -> "reduction in T. b. gambiense case numbers".

"Finally, abundance of PAD proteins" - should be "the abundances of transcripts encoding PAD proteins..." - plural needed, and proteins were not measured.

Line 271: "|the cpsf3 gene" Gene should be in capital letters and italics.

"in clinical practise as.." -> "practice" ("Practise" is a verb.)

Reviewer #2: My view is that operarational this is a major 'minor revision'.

Reviewer #3: There are some minor changes required:

Line 70: Only eflornithine is biologically species specific, it is policy that dictates the use of the other drugs, for example suramin is not used because of coinfection with filarial nematodes, not because T. gambiense is resistant to it. Also, the word disease should be inserted in front of stage-specific to distinguish it from parasite stages.

Line 117: Lister 427 is a monomorphic strain with a defective differentiation pattern - authors should comment on this in the discussion, in particular as regards mitochondrial activation for example.

Line 123 numbers should be given for doubling times.

Line 267 "step chance" should be "step change"

Line 284 insert the word partial before differentiation as there was no morphological differentiation.

Lines 424-426 Fig 1. Legend to part B does not refer to the figure but talks about changes in EC50 whereas the figure is about doubling times.

Line 426 The C is missing where the legend should refer to part C.

**Summary and General Comments**

Reviewer #1: See above for overall comments. The study is interesting but without more detailed analysis of the transcriptomes and genotype, including effects of removing the selection, the results are difficult to interpret. If the cells are resistant simply because they are growing more slowly, the reasons for the slow growth need to be examined.

Reviewer #2: The authors describe the generation and characterization of an acoziborole-resistant bloodstream Trypanosoma brucei cell line. They report co-resistance to sinefungin and hypersensitivity to other drugs. Overall, it’s a straightforward manuscript, but one that I enjoyed reading and the work will be of value and interest to others in the community. However, as listed below, there are areas that need tidying before the work is suitable for publication.

Line 123, cell doubling times lengthen (i.e. increase) not reduced (Fig 1B). In the legend to Fig 1B, the growth curve is not described – rather B) in the legend is referring to Fig 1C. And finally with respect to Fig 1B the cell densities per ml seem rather high (and don’t tally with what’s in the Materials and Methods for in vitro culture. Or am I looking at an in vivo growth curve? If in vitro are the resistant clones grown up in the presence of drug or not? None of the cultures, if these are batch cultures, appear to have reached stationary phase – this is particularly of note for the resistant cultures since at what density do they plateau out?

More generally, 4 clones from the same resistant population are analysed, so the wider question is how representative is the resistance mechanism described in the current work? Of course, were a second resistant population analysed and it presented a different resistance phenotype, one would be content with reading the description of two different resistance phenotypes – so I have no problem with the single resistant population analysed but

I see no indication that expression, at the protein level, of the expressed VSG is reduced, but much appears made of the changes in VSG expression seen at the level of transcriptomics. Is the 221 VSG actually translated at a lower level in the resistant population versus the parent? Unless the authors are clearer on the effect of resistance on VSG expression, the reader is left hanging – I’m not sure for instance that if say endo/exocytic rate were reduced this wouldn’t affect the amount of VSG on the actual cell surface.

Did the authors complete any TEM – is the classic appearance of the VSG coat (e.g. as noted by Vickerman) evident all around the cell or is that surface coat now patchy or the cell surface changed in some other way evident in EM?

Table 2 – would be helpful to know by reading (rather than dig through TriTrypDB) whether the several hypotheticals are T. brucei-specific trypanosomatid/kinetoplastid-specific, chromosome-internal or sub-telomeric. More details please.

At the level of proteomics or immunoblotting is there any evidence for translation of procyclic marker proteins e.g. PPDK, PARP, transporters in the resistant cell line – the authors talk about a ‘differentiation event’ (or phenotype) (line 284) but I don’t see any evidence really for anything other than a bloodstream cell with an altered transcriptome and odd kintoplast-nucleus distance and the latter perhaps is a function of slow cell growth.

How long did the authors growth the resistant mutants for – does the growth rate ever speed up? Is the resistance phenotype (or slow growth) retained after drug pressure is removed? These details were also missing but would be helpful to know.

Reviewer #3: This study provides an interesting insight into novel mechanism of drug resistance in African trypanosomes and also provides new information about the differentiation process. It is interesting that the procyclin genses are upregulated as well as other PCF specific genes, since the procyclins are transcribed from specialise dloci by RNA polymerase I like the VSG genes so the transcriptional switch posited by the authors must affect the RNA Pol I stage regulated system as well as RNA Pol II transcribed genes. This resistance mechanism is unlikely to operate easily in the field as PCF-like forms are unlikely to survive in an immunocompetent mammalian host as the authors state.

There are two major weaknesses, one is that the authors have not demonstrated that any of the upregulation at the RNA level leads to upregulation of the encoded proteins, trypanosomes display translational control so it is not automatic that increased mRNA leads to increased protein levels. The authors should mention this caveat in their discussion. The second is that they have identifed a phenomenon - "transcriptional differentiation" which accompanes the acquisition of acoziborole resistance but they have not identifed a mechanism whereby this leads to resistance. 

Overall the study is well-performed within the above limitations and is both novel and significant in increasing our understanding of the plasticity of trypanosome phenotypes.

PLOS authors have the option to publish the peer review history of their article (what does this mean?). If published, this will include your full peer review and any attached files.

Reviewer #1: Yes: Christine Clayton

Reviewer #2: No

Reviewer #3: No
---

## [Decision Letter · Decision Letter 1]

13 Oct 2021

Dear Prof. Barrett,

Thank you very much for submitting your manuscript "Transcriptional differentiation of Trypanosoma brucei during in vitro acquisition of resistance to acoziborole" for consideration at PLOS Neglected Tropical Diseases. 

Your manuscript has been re-reviewed by the original three reviewers. I'm happy to tell you that they feel you have addressed their concerns and that they all recommend publication. Reviewer 1 has a few minor edits that they would like to see incorporated so I am sending the manuscript back to you so you can consider those recommendations. I shall be happy to accept your manuscript once you have had an opportunity to address those recommendations.

Sincerely,

Margaret A Phillips, Ph.D.

Deputy Editor

Margaret Phillips

Deputy Editor

Your manuscript has been re-reviewed by the original three reviewers. I'm happy to tell you that they feel you have addressed their concerns and that they all recommend publication. Reviewer 1 has a few minor edits that they would like to see incorporated so I am sending the manuscript back to you so you can consider those recommendations. I shall be happy to accept your manuscript once you have had an opportunity to address those recommendations.

Reviewer's Responses to Questions

**Key Review Criteria Required for Acceptance?**

**Methods**

-Are the objectives of the study clearly articulated with a clear testable hypothesis stated?

-Is the study design appropriate to address the stated objectives?

-Is the population clearly described and appropriate for the hypothesis being tested?

-Is the sample size sufficient to ensure adequate power to address the hypothesis being tested?

-Were correct statistical analysis used to support conclusions?

-Are there concerns about ethical or regulatory requirements being met?

Reviewer #1: yes

Reviewer #2: Yes, to all criteria requested.

Reviewer #3: The revised version is accepatble

**Results**

-Does the analysis presented match the analysis plan?

-Are the results clearly and completely presented?

-Are the figures (Tables, Images) of sufficient quality for clarity?

Reviewer #1: all OK

Reviewer #2: Yes, to all criteria requested.

Reviewer #3: The revised version is acceptable

**Conclusions**

-Are the conclusions supported by the data presented?

-Are the limitations of analysis clearly described?

-Do the authors discuss how these data can be helpful to advance our understanding of the topic under study?

-Is public health relevance addressed?

Reviewer #1: Yes now all OK

Reviewer #2: Yes, to all criteria requested.

Reviewer #3: The manuscript is improved by the authors responses to the reviewers and is acceptable for publication in my opinion.

**Editorial and Data Presentation Modifications?**

Reviewer #1: no

Reviewer #2: (No Response)

Reviewer #3: Accept

**Summary and General Comments**

Reviewer #1: This paper is very much improved. There are only a few minor things now, which have resulted from new introductions to the text.

I noted that the authors talk about procyclic and stumpy in the Introduction but don’t actually say what stumpy forms are, which would be a problem for potential readers who don’t work on trypanosomes but are interested in the benzoxoboroles. A couple of sentences that say what stumpy forms are and list genes important for (and markers for) stumpy differentiation are needed. Aslo somewhere the authors need to describe the morphological differences between the three forms considered.

ine 116 _ change to “a partial switch towards procyclic mRNA abundances”. I find the word “phenotype “ a bit too strong. Many readers might assume that “phenotype” means morphology, which was not much changed.

Line 1272 - “transcript” not “transcription”.

“Transcripts associated with proteins involved in stage-specific regulation of mRNA (e.g. reduction in PARN-1; Tb927.8.2850) were also significantly altered. “ The evidence that PARN1 is involved in stage-specific regulation of mRNA is based on artificial over-expression. The effects on differentiation have not been measured because the only paper (Utter et al, not cited) is with non-differentiation-competent cells. A procyclic knock-out grew almost normally and RNAi had no effect in bloodstream forms. Actually, looking at the resutls of the microarrays, they are really wierd -a whole cluster from one particular region of chromosome 9 is all up about 2-fold. It’s inducible so unlikely to be a gene duplication. The only genes affected that are linked to differentiation are the BARP genes, which are actually a marker for some salivary gland epimastigotes. in that context, there’s an increase in HAP2 mRNA in the acoziborole resistant cells, which suggests to me a stress response since it’s a gamete protein. No harm in mentioning PARN1 but I’d be careful about saying what it does. If you want to find proteins implicated in control of gene expression in the bloodstream-stumpy-procyclic transition you would be on more solid ground with the known markers and signalling proteins, and with ZC3H11 and ZC3H22 (Tb927.7.2680).

 I was intrigued by the set of proteins that are affected in the slender-vs-stumpy or slender -vs-procyclic comparisons, but not in the acoziborole-resistant cells. For stumpy forms they seem to be dominated by cytoskeletal proteins and histones - the stumpy cells aren’t dividing and the resistant cells are.

Reviewer #2: (No Response)

Reviewer #3: The manuscript is improved by the authors responses to the reviewers and is acceptable for publication in my opinion.

PLOS authors have the option to publish the peer review history of their article (what does this mean?). If published, this will include your full peer review and any attached files.

Reviewer #1: Yes: Christine Clayton

Reviewer #2: No

Reviewer #3: No

Figure Files:

Data Requirements:

Reproducibility:

References

---

## [Editor Report · Decision Letter 2]

21 Oct 2021

Dear Prof. Barrett,

We are pleased to inform you that your manuscript 'Transcriptional differentiation of Trypanosoma brucei during in vitro acquisition of resistance to acoziborole' has been provisionally accepted for publication in PLOS Neglected Tropical Diseases.

Best regards,

Margaret A Phillips, Ph.D.

Deputy Editor

Margaret Phillips

Deputy Editor

---

## [Editor Report · Acceptance letter]

5 Nov 2021

Dear Prof. Barrett,

We are delighted to inform you that your manuscript, "Transcriptional differentiation of Trypanosoma brucei during in vitro acquisition of resistance to acoziborole," has been formally accepted for publication in PLOS Neglected Tropical Diseases.

Best regards,

Shaden Kamhawi

co-Editor-in-Chief

Paul Brindley

co-Editor-in-Chief
